# Cardiovascular and Respiratory Effects of Increased Intra-Abdominal Pressure with and without Dexmedetomidine in Anesthetized Dogs

**DOI:** 10.3390/vetsci10110634

**Published:** 2023-10-27

**Authors:** Dongseok Kim, Minjun Seo, Geonho Choi, Sang-Kwon Lee, Sungin Lee, Won-Jae Lee, Sung-Ho Yun, Young-Sam Kwon, Min Jang

**Affiliations:** 1Department of Veterinary Surgery, College of Veterinary Medicine, Kyungpook National University, Daegu 41566, Republic of Korea; 20105asd@knu.ac.kr (D.K.); rjshrjsh321@gmail.com (G.C.); sklee10@knu.ac.kr (S.-K.L.); iamcyshd@knu.ac.kr (W.-J.L.); shyun@knu.ac.kr (S.-H.Y.); kwon@knu.ac.kr (Y.-S.K.); 2Department of Veterinary Surgery, College of Veterinary Medicine, Chungbuk National University, Cheongju 28644, Republic of Korea; sunginlee@cbnu.ac.kr

**Keywords:** dog, intra-abdominal pressure, dexmedetomidine, hypercapnia, PulseCO

## Abstract

**Simple Summary:**

Laparoscopic procedures have been gaining popularity in veterinary medicine. However, there are no studies evaluating the safety of using dexmedetomidine as an adjuvant during laparoscopy in the veterinary literature. High intra-abdominal pressure and dexmedetomidine might have deleterious cardiorespiratory effects; therefore, this study was conducted to evaluate the effects of intra-abdominal pressure and dexmedetomidine on cardiorespiratory variables in healthy dogs. Five healthy beagle dogs were enrolled in the study, which was conducted with a crossover design. Cardiovascular and respiratory variables were monitored at different intra-abdominal pressures through inducing capnopertioneum. After a washout period, the same protocols were applied with dexmedetomidine administration. Our study revealed that no significant cardiorespiratory effects were observed until intra-abdominal pressure reached 20 mmHg, as well as during the administration of dexmedetomidine. These findings have shown that the administration of a dexmedetomidine infusion may be applicable in laparoscopic procedures in healthy dogs.

**Abstract:**

Intra-abdominal pressure (IAP) elevation during capnoperitoneum can cause adverse cardiovascular and respiratory effects. This study aimed to determine if a sequentially increased IAP affects cardiovascular and respiratory variables in anesthetized dogs and evaluate the effects of the constant-rate infusion of dexmedetomidine (Dex) on cardiovascular and respiratory variables with increased IAP. Five dogs were anesthetized and instrumented, and a Veress needle was equipped to adjust the IAP using a carbon dioxide insufflator. Stabilization was conducted for 1 h, and physiological variables were measured at IAPs of 0, 5, 10, 15, and 20 mmHg and after desufflation. After the washout period, the dogs underwent similar procedures along with a constant-rate infusion of dexmedetomidine. The cardiovascular effects of increased IAP up to 20 mmHg were not significant in healthy beagle dogs and those administered with dexmedetomidine. When comparing the control and dexmedetomidine groups, the overall significant effects of dexmedetomidine were noted on heart rate, cardiac output, and systemic vascular resistance during the experiment. Respiratory effects were not observed during abdominal insufflation when compared between different IAPs and between the two groups. Overall, an increased IAP of up to 20 mmHg did not significantly affect cardiovascular and respiratory variables in both the control and dexmedetomidine groups. This study suggests that the administration of a dexmedetomidine infusion is applicable in laparoscopic procedures in healthy dogs.

## 1. Introduction

The abdominal cavity is a relatively compliant confined space, which allows a relatively limited capacity to increase the intracavitary volume or reduce extracavitary compliance before a relative increase in intra-abdominal pressure (IAP) occurs [1]. Increased IAP occurs in several clinical situations, including bandaging, obesity, intra-abdominal visceral content distension, and capnoperitoneum for laparoscopy [1]. Among these, as carbon dioxide insufflation for the laparoscopy approach is gaining popularity, clinical and experimental studies have shown that capnoperitoneum can cause adverse cardiovascular and respiratory effects [2,3]. To minimize these effects, maintaining the insufflation pressure below 12 mmHg in dogs has been recommended [4].

Dexmedetomidine (Dex), an alpha-2 adrenergic agonist, is used to provide reliable sedation, analgesia, and chemical restraint in dogs [5]. Moreover, it has properties that reduce the doses of induction and maintenance drugs and dose-dependently decrease the minimum alveolar concentration (MAC) of isoflurane in dogs [6,7]. Furthermore, these alpha-2 agonists generally have significant effects on cardiovascular function, including bradycardia, increased systemic vascular resistance (SVR), and decreased cardiac output (CO) [8]. In human medicine, dexmedetomidine is frequently used as an adjuvant to general anesthesia in the laparoscopy procedure [9,10,11]. However, in veterinary medicine, the hemodynamic effects of dexmedetomidine during abdominal insufflation with CO_2_ have not been thoroughly investigated.

CO is frequently measured during anesthesia or in critical care medicine, allowing for the early detection of hemodynamic instability [12]. In veterinary medicine, various techniques have been evaluated for CO measurement, with the pulmonary arterial catheter thermodilution (PAC-TD) method being the gold standard [13]. However, due to the serious complications associated with these methods, including arrhythmias, thrombosis, and infections, less invasive alternatives have been explored to mitigate these risks [14]. One such alternative is the PulseCO method, which calculates CO through pulse contour analysis. This method offers the advantage of real-time CO monitoring, enabling clinicians to promptly assess a patient’s response to treatment [15]. In this study, we monitored CO using the PulseCO method with a LiDCO™ hemodynamic monitor (LiDCO Unity; Masimo, CA, USA).

Recently, minimally invasive surgeries using endoscopic procedures have been gaining popularity in veterinary medicine [16]. Although several studies have investigated the cardiovascular and respiratory effects of abdominal insufflation in anesthetized dogs, to the best of our knowledge, none have monitored parameters with a constant-rate infusion of dexmedetomidine during abdominal insufflation. Therefore, this study aimed to determine whether sequentially increased IAP affects cardiovascular and respiratory variables in anesthetized dogs and evaluate the effects of a constant-rate infusion of dexmedetomidine on these variables in the presence of increased IAP.

## 2. Materials and Methods

### 2.1. Animals

Five young adult beagles (three females and two males) were included in this study after obtaining approval from the Institutional Animal Care and Use Committee of Kyungpook National University (approval number: KNU2022-0348). They were 10–14 months old, with a median body weight of 10 (range, 9.4–10.5) kg. Prior to the study, all dogs underwent physical examinations, blood analyses, and radiography, and they were deemed to be in good health. The body condition score was 5 (range, 4–6) points on a 9-point scale. The study followed a prospective crossover design with a 7-day washout period between treatments.

### 2.2. Anesthesia and Instrumentation

The dogs underwent an 8 h fasting period, with access to water until they were transferred to the examination room for anesthesia. A 22-gauge catheter was then placed in one of the cephalic veins, and crystalloid fluids (Hartmann solution; JW Pharmaceutical, Seoul, Korea) were administered at a rate of 5 mL/kg/h. Before anesthesia, the dogs were premedicated with tramadol (Maritrol; Jeil Pharmaceutical Co., Ltd., Daegu, Korea) at a dose of 4 mg/kg (intramuscularly (IM)) and cefazoline (Cefozol; Hankook Korus Pharm, Seoul, Korea) at a dose of 20 mg/kg (intravenously (IV)). Preoxygenation involved administering 100% oxygen at a flow rate of 5 L/min for 5 min, followed by anesthetic induction using alfaxalone (Alfaxan Multidose; Jurox Pty, Maitland, Australia) at a dose of 2 mg/kg (IV) over 1 min. After endotracheal intubation, each dog was positioned in dorsal recumbency, and anesthesia was maintained using isoflurane (Isoflurane; Hana Pharm, Seoul, Korea) in 100% oxygen through a circle rebreathing system (9100c NXT; GE Healthcare, Wuxi, China).

Each dog was equipped with monitoring instruments to measure physiological variables. We used a multiparameter monitor (CARESCAPE Monitor B650; GE Healthcare, Helsinki, Finland) to track heart rate (HR, via lead II electrocardiogram), oxygen saturation (SpO_2_, via pulse oximetry), esophageal temperature, and systolic, diastolic, and mean arterial blood pressures (SAP, DAP, and MAP, respectively). For arterial blood pressure measurement and blood collection for arterial blood gas analysis, we placed an over-the-needle 22-gauge catheter in the dorsal pedal artery. Arterial blood samples were collected and analyzed using a blood gas analyzer (ABL80 FLEX; Radiometer, Bronsjoj, Denmark) to determine the pH, partial pressure of oxygen in arterial blood (PaO_2_), partial pressure of carbon dioxide in arterial blood (PaCO_2_), bicarbonate (HCO_3_^−^), and base excess (BE). Dorsal pedal artery catheters were connected to a pressure transducer (TruWave; Edwards Lifesciences, Munich, Germany), positioned at the level of the heart base, and connected to a catheter using a noncompliant tubing system continuously flushed with heparinized saline (0.9% NaCl; JW Pharmaceutical, Seoul, Korea). Arterial pulse pressure data, analyzed in the transducer, were transmitted from the multiparameter monitor to the LiDCO™ monitor. The LiDCO™ monitor employs the PulseCO method, not the LiDCO system, to continuously calculate stroke volume (SV), CO, and SVR data in real-time. We also measured end-tidal carbon dioxide (ETCO_2_), the respiratory rate (RR), and the expired fraction of isoflurane (FE’Iso) using a side-stream infrared gas analyzer (CARESCAPE Monitor B650; GE Healthcare, Finland). Tidal volume (TV) and minute volume (MV) were recorded using a volume monitor (9100c NXT; GE Healthcare, Wuxi, China). If ETCO_2_ exceeded 60 mmHg, intermittent positive pressure ventilation (IPPV) was initiated.

### 2.3. LiDCO™ Monitor Calibration

After stabilization, an echocardiogram was performed using an ultrasound machine (Versana Active; GE Healthcare, Helsinki, Finland) to determine the CO and calibrate the LiDCO™ monitor. In the present study, the velocity–time integral method was applied to measure the CO. This method was performed through measuring the velocity of the left ventricular outflow tract in the left apical five-chamber view using pulsed-wave Doppler. Subsequently, the aortic diameter was measured in the right parasternal long-axis view. SV was calculated using the formula π × (aortic diameter/2)^2^, and HR was multiplied to obtain the CO. The CO of each dog was recorded as the mean of three consecutive measurements.

### 2.4. Veress Needle Placement

The ventral region of the abdomen was clipped and aseptically prepared. Subsequently, a skin incision was made 20 mm caudal to the umbilicus. A 14-gauge Veress needle (Veress needle, ADDLER, Mumbai, India) was inserted in the incision site, and the correct needle placement was checked via the injection of 5 mL normal saline, followed by aspiration. The Veress needle was connected to an automatic CO_2_ insufflator (UHI-3; Olympus Optical Inc., Tokyo, Japan), which can adjust the IAP.

### 2.5. Experimental Design

The instrumentation period was standardized to 60 min. Subsequently, FE’Iso was maintained at 1.73% (1.3 × MAC) [17], and the dogs were allowed to breathe spontaneously. Thereafter, baseline data of the following physiologic variables were recorded: HR, SpO_2_, esophageal temperature, SAP, DAP, MAP, EtCO_2_, RR, TV, MV, pH, PaO_2_, PaCO_2_, HCO_3_^−^, BE, SV, CO, and SVR. After data collection, the IAP was gradually increased to 5, 10, 15, and 20 mmHg through insufflating CO_2_. The dogs were allowed to stabilize for 10 min between each of the IAP adjustments. Physiological variables were measured at six time points: T0 (baseline), immediately after the instrumentation period; T1, 10 min after adjusting IAP to 5 mmHg; T2, 10 min after adjusting IAP to 10 mmHg; T3, 10 min after adjusting IAP to 15 mmHg; T4, 10 min after adjusting IAP to 20 mmHg; and T5, 10 min after desufflation. Following desufflation, meloxicam (0.2 mg/kg, subcutaneously [SC]) was administered to all dogs before their recovery from anesthesia. All catheters were removed, and the dogs were returned to their housing facility.

After a 1-week washout period, the dogs underwent similar procedures until 10 min before T0, and an IV loading dose of dexmedetomidine (3 µg/kg) was administered followed by an infusion of 3 µg/kg/h of dexmedetomidine. Immediately following the administration, the FE’Iso was decreased to 0.72% based on a previous study [6] to maintain isoflurane–dexmedetomidine-administered dogs equipotent to 1.3 MAC isoflurane-administered-alone dogs. Subsequently, measurements of baseline variables at T0 were recorded, and a similar experimental protocol was performed with infusions of dexmedetomidine. The study timeline is summarized in Figure 1. The dogs that were not administered dexmedetomidine were assigned as the control group and those which were administered dexmedetomidine were assigned as the Dex group.

### 2.6. Statistical Analysis

The normal distribution of the data was confirmed using the Shapiro–Wilk test before analysis. All values were presented as means and standard deviations. To compare each physiological variable within the two groups, a repeated-measures one-way analysis of variance was performed. Additionally, the Kruskal–Wallis test was performed for nonnormally distributed data. If significant differences were identified, a post hoc analysis was performed using the Bonferroni test. A Wilcoxon signed-rank test or Mann–Whitney U test was performed for each variable between the two groups at each time point. A *p* value of <0.05 was considered statistically significant. Analyses were performed using Statistical Package for the Social Sciences for Windows (version 25, IBM, Armonk, NY, USA).

## 3. Results

All five dogs completed the study without any complications. CO measurements using echocardiograms of the five dogs ranged from 1.264 to 1.75 (mean, 1.47) L/min. No significant changes were observed in body temperature (35.1–37.9 °C) and oxygen saturation (94–100%) over time and between treatments.

Table 1 summarizes the cardiovascular changes resulting from increased IAP with carbon dioxide and dexmedetomidine administration. No significant differences were observed in variables between different IAPs in both groups. In the control group, HR, MAP, SAP, and DAP showed an increase in T3 and T4 compared with T0; however, the results were not statistically significant (HR, *p* = 1.000, *p* = 1.000; MAP, *p* = 1.000, *p* = 0.399; SAP, *p* = 1.000, *p* = 0.293; and DAP, *p* = 0.825, *p* = 0.141; for T3 and T4 compared with T0, respectively). Compared with the control group, overall significant effects of dexmedetomidine were observed on HR, CO, and SVR during the experiment. SAP, MAP, and DAP increased in T0, T1, T2, and T5 with dexmedetomidine treatment, whereas no differences were noted in T3 and T4 between the control and Dex groups. In the Dex group, SV values decreased significantly in T0, T1, and T2.

The effects of intra-abdominal insufflation with carbon dioxide on respiratory and blood gas variables in the control and Dex groups are summarized in Table 2. No significant differences in EtCO_2_, RR, TV, MV, pH, PaO_2_, PaCO_2_, HCO_3_^−^, and BE were observed between different IAPs and between the two groups. In both treatments, PaCO_2_ was increased following carbon dioxide insufflation compared with that at baseline T0; however, the results were not statistically significant. Moderate-to-severe hypercapnia (EtCO_2_ > 50 mmHg) was detected in all five dogs. One dog showed an EtCO_2_ of >60 mmHg in both treatment periods, and IPPV could not resolve the hypercapnia.

## 4. Discussion

Previous studies have shown that intra-abdominal hypertension not only affects cardiorespiratory function but also disrupts regional blood flow and impairs tissue perfusion, potentially leading to organ dysfunction [1,4]. However, since our primary objective was to evaluate cardiovascular and respiratory effects, we focused solely on these aspects. Therefore, this study compared cardiovascular and respiratory variables in anesthetized dogs among different sequentially increased IAPs and between groups that received or did not receive dexmedetomidine. In general, an increased IAP of up to 20 mmHg did not significantly affect cardiovascular and respiratory variables in the control group, which did not receive premedication. In the Dex group, we compared hemodynamic changes with those in the control group. However, no significant cardiovascular or respiratory alterations were observed during abdominal insufflation.

The cardiovascular effects of increased IAP result from the cranial displacement of the diaphragm and compression of the caudal vena cava, directly compressing the heart and leading to reduced preload and CO [1]. In the control group of our study, HR and arterial pressure showed a slight but statistically insignificant increase when IAP was raised to 15 and 20 mmHg. The modest increase in arterial pressure during insufflation may have been due to an increase in SVR and a decrease in CO and SV. These findings align with previous reports that noted decreased CO and increased SVR in response to elevated IAP [18,19,20]. However, those studies typically involved higher pressures (>20 mmHg). For instance, Duke et al. investigated lower IAPs (15 mmHg) and found increased SVR with maintained CO [21]. While the maintenance of CO in that study may be attributed to the lower IAP, our study observed a slight decrease in CO at an IAP of 15 mmHg. These results indicate that IAPs higher than 15 mmHg may contribute to changes in cardiovascular variables. Our study also revealed the cardiovascular effects of dexmedetomidine, with the dexmedetomidine group showing increased arterial blood pressure and decreased HR. These findings align with other studies assessing the cardiovascular effects of alpha-2 agonists, which are believed to be a result of the baroreceptor-mediated response—a widely accepted side effect of alpha-2 agonists [5,6,8].

As previously mentioned, both alpha-2 agonists and abdominal insufflation can impact the cardiovascular system. In human medicine, several studies have evaluated the effects of dexmedetomidine on hemodynamic variables in patients undergoing laparoscopic procedures. These studies used dexmedetomidine infusion rates of 0.1 and 0.2 µg/kg/h, respectively, which provided stable hemodynamic conditions during laparoscopic surgery [9,10]. In our study, we administered a dexmedetomidine loading dose of 3 µg/kg, followed by an infusion rate of 3 µg/kg/h. The results showed that in the Dex group, cardiovascular variables during abdominal insufflation (T1–5) did not significantly differ from those at baseline (T0). These findings suggest that hemodynamic stability was observed in anesthetized dogs administered 3 µg/kg/h of dexmedetomidine during abdominal insufflation up to 20 mmHg. However, it is important to note that our study used only one dexmedetomidine dose. To assess the effects of dexmedetomidine on hemodynamic status in laparoscopic procedures with various dexmedetomidine doses, further research is warranted.

In this study, we measured CO, SV, and SVR using the LiDCO™ monitor, which calculates CO through pulse contour analysis. Previous studies have assessed the PulseCO method and reported positive results, demonstrating correlations with the PAC-TD method, considered the gold standard [13,22,23]. While the PulseCO method was originally developed for human medicine, it requires a one-point calibration. These studies conducted calibration using CO measurements obtained from the lithium dilution or thermodilution method [13,22,23]. In contrast, our study employed an innovative approach, calibrating the monitor with CO measurements obtained through echocardiography. This calibration method maximizes the advantages of being less invasive, considering the relative invasiveness and associated risks of complications with the lithium and thermodilution methods. Additionally, the real-time monitoring of CO trends provided by the PulseCO method is clinically essential for patient care [13]. However, it is important to note that the PulseCO method also has its limitations. One study comparing CO measurements between echocardiography and the thermodilution method reported that echocardiographic methods tended to underestimate CO [24]. Furthermore, echocardiogram artifacts have been identified as potential causes of CO underestimation [24]. Moreover, the CO measurements obtained using the LiDCO™ monitor in our study may have some degree of underestimation; therefore, further research is necessary to establish the reliability of this novel technique.

Previous studies have demonstrated that respiratory disturbances associated with elevated IAPs can occur in pediatric human patients as well as animals [20,25]. These effects are thought to be triggered by the cranial displacement of the diaphragm, leading to a reduction in functional residual capacity, residual volume, and total lung capacity [1]. Richardson et al. indicated that significant respiratory effects were observed at IAPs exceeding 25 mmHg, while our findings similarly indicate that no significant differences in respiratory variables were observed until IAP reached 20 mmHg [20]. However, the respiratory disturbance identified in the present study was that all dogs exhibited moderate-to-severe hypercapnia (EtCO_2_ > 50 mmHg) throughout the anesthesia period, regardless of different IAPs. While the causes of hypercapnia can be diverse, in this study, it is believed that respiratory suppression due to anesthetics and the elimination of carbon dioxide through the lungs during capnoperitoneum were the main factors leading to hypercapnia, rather than increased IAP [26,27,28]. This is supported by a previous study conducted on conscious dogs, which showed that hypercapnia was not observed throughout the experiments [29]. In conclusion, maintaining normocapnia and preventing respiratory acidosis is recommended through adequate ventilation during laparoscopic procedures.

In one dog, in the current study, hypercapnia (EtCO_2_ > 50 mmHg) was observed before abdominal insufflation; after IAP insufflation of up to 15–20 mmHg, end-tidal carbon dioxide concentration reached more than 75 mmHg. Despite persistent manual positive pressure ventilation, severe hypercapnia did not resolve. After desufflation, EtCO_2_ levels decreased to the level before insufflation. However, why this dog exhibited severe hypercapnia remains unclear.

The present study had several limitations. First, this study was conducted with a small sample size. To minimize the influence of individual animal effects on physiological variables, the dogs used in this study were of the same breed, similar size, and age, and a crossover design was applied. Second, the CO_2_ insufflator has an automatic relief function that automatically releases excess pressure when the pressure exceeds 20 mmHg. Thus, our study could not evaluate the effects of IAPs over 20 mmHg. Third, plasma drug concentrations were not measured; therefore, possible variations in concentration and their influence on circulation cannot be assessed within the present study. Lastly, the body positions of the dogs were not considered in the present study. Laparoscopic procedures are commonly performed in different positions, such as inverse or reverse Trendelenburg positions; as these positions influence cardiovascular variables, further study is needed to evaluate physiological parameters in different positions.

## 5. Conclusions

An increased IAP of up to 20 mmHg did not have any significant cardiovascular and respiratory effects in healthy beagle dogs. Furthermore, the dogs which were administered dexmedetomidine were not affected by abdominal insufflation. Therefore, this study demonstrates that the administration of a dexmedetomidine infusion may be applicable in laparoscopic procedures in healthy dogs. However, further studies are warranted as the present study was conducted in only one body position and had limited IAPs evaluated.

## Figures and Tables

**Figure 1 vetsci-10-00634-f001:**
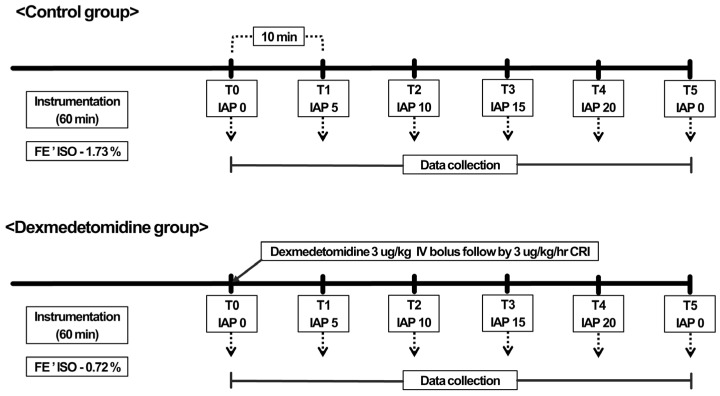
Graphical representation of the study timeline.

**Table 1 vetsci-10-00634-t001:** Mean and standard deviation of cardiovascular variables across different IAPs between the control and dexmedetomidine groups.

		T0 (IAP 0 mmHg)		T1 (IAP 5 mmHg)		T2 (IAP 10 mmHg)		T3 (IAP 15 mmHg)		T4 (IAP 20 mmHg)		T5 (IAP 0 mmHg)	
		Mean	SD	Mean	SD	Mean	SD	Mean	SD	Mean	SD	Mean	SD
HR (/min)	Control	103.0	13.7	111.2	13.5	110.8	15.9	114.8	11.9	116.0	14.4	117.0	14.6
	Dex	81.0 ^†^	11.0	71.2 ^†^	10.0	70.0 ^†^	17.7	69.8 ^†^	9.2	67.8 ^†^	9.09	64.8 ^†^	4.8
SAP (mmHg)	Control	97.0	18.9	94.4	9.0	91.2	7.6	112.6	16.4	122.6	24.6	95.0	14.3
	Dex	154.2 ^†^	14.0	146.0 ^†^	11.1	140.8 ^†^	10.4	134.6	8.7	135.2	11.0	131.4 ^†^	12.1
MAP (mmHg)	Control	68.4	17.5	68.0	8.2	66.6	7.7	84.8	15.1	90.8	22.2	67.4	13.9
	Dex	124.0 ^†^	11.8	114.0 ^†^	9.8	103.8 ^†^	14.7	98.4	8.6	97.8	13.1	96.8 ^†^	8.2
DAP (mmHg)	Control	53.6	17.6	53.6	9.8	53.0	8.6	72.6	16.0	80.2	20.6	56.4	12.9
	Dex	111.0 ^†^	11.1	98.4 ^†^	12.5	88.6 ^†^	13.2	84.2	9.2	84.0	14.1	83.0 ^†^	10.0
CO (L/min)	Control	1.5	0.4	1.5	0.4	1.5	0.4	1.3	0.3	1.2	0.4	1.6	0.3
	Dex	0.7 ^†^	0.2	0.6 ^†^	0.2	0.7 ^†^	0.2	0.8 ^†^	0.2	0.6 ^†^	0.2	0.7 ^†^	0.1
SV (mL)	Control	13.7	2.7	13.7	2.5	13.0	2.6	11.1	2.1	11.1	2.1	13.4	2.7
	Dex	7.8 ^†^	1.4	8.2 ^†^	1.6	9.1 ^†^	1.9	9.8	1.6	10.1	1.4	10.4	1.0
SVR (Dynes/ s/cm^−5^)	Control	3621.1	1416.7	3467.5	1272.1	3563.2	1267.3	5043.2	1972.0	5962.9	2286.7	3436.2	1302.9
	Dex	17,577.4 ^†^	5397.8	18,031.9 ^†^	8199.3	13,097.5 ^†^	3921.4	10,535.2 ^†^	2745.0	12,548.2 ^†^	4613.3	11,417.9 ^†^	2510.2

IAP, intra-abdominal pressure; HR, heart rate; SAP, systolic arterial blood pressure; MAP, mean arterial blood pressure; DAP, diastolic arterial blood pressure; CO, cardiac output; SV, stroke volume; SVR, systemic vascular resistance; SD, standard deviation. Significantly different from the baseline (T0) within the group (*p* < 0.05); however, no differences are observed in cardiovascular variables between different time points. ^†^ Significantly different from the control group at the same time point in the dexmedetomidine group (*p* < 0.05).

**Table 2 vetsci-10-00634-t002:** Mean and standard deviation of respiratory and blood gas variables across different IAPs between the control and dexmedetomidine groups.

		T0 (IAP 0 mmHg)		T1 (IAP 5 mmHg)		T2 (IAP 10 mmHg)		T3 (IAP 15 mmHg)		T4 (IAP 20 mmHg)		T5 (IAP 0 mmHg)	
		Mean	SD	Mean	SD	Mean	SD	Mean	SD	Mean	SD	Mean	SD
RR (/min)	Control	13.2	6.3	14.6	5.9	17.4	7.8	17.2	5.4	20.2	6.5	17.4	4.5
	Dex	12.4	2.7	14.2	4.2	18.8	6.8	21.0	7.2	21.0	6.9	15.4	2.1
EtCO_2_ (mmHg)	Control	55.4	5.4	58.0	7.3	56.0	6.5	59.2	8.6	56.4	11.5	53.4	7.0
	Dex	51.0	4.5	50.6	5.0	51.0	5.2	53.6	7.0	53.2	8.4	44.8	3.0
TV (ml/min)	Control	139.6	27.8	136.0	24.9	136.8	32.4	129.6	14.5	125.6	17.5	155.6	32.1
	Dex	146.0	17.7	155.6	6.6	130.8	4.9	117.8	10.2	113.4	12.2	158.8	10.9
MV (L/min)	Control	1.9	0.7	1.7	0.8	2.2	0.6	2.0	0.8	2.5	1.1	2.7	0.5
	Dex	1.9	0.6	2.1	0.7	2.6	1.0	2.4	1.0	2.4	0.9	2.5	0.5
pH	Control	7.4	0.1	7.3	0.0	7.3	0.1	7.3	0.1	7.3	0.1	7.3	0.1
	Dex	7.4	0.1	7.3	0.1	7.3	0.1	7.3	0.1	7.3	0.1	7.4	0.1
PaO_2_ (mmHg)	Control	447.4	60.9	472.6	70.5	477.2	113.4	503.6	73.7	516.8	93.6	508.6	81.1
	Dex	481.2	62.3	520.2	29.9	521.2	48.9	522.6	25.7	525.0	56.4	538.8	53.9
PaCO_2_ (mmHg)	Control	53.6	6.8	56.9	9.0	56.3	8.5	59.9	9.5	57.4	7.0	54.7	8.2
	Dex	47.6	5.1	52.0	4.4	54.6	5.6	53.9	5.9	56.4	10.0	51.1	8.8
HCO_3_^−^ (mmol/L)	Control	30.1	2.9	29.4	2.7	29.5	3.0	28.9	4.3	28.5	4.4	27.8	3.6
	Dex	26.5	4.9	26.8	4.9	27.4	5.4	27.0	5.8	28.1	6.0	28.2	5.0
BE (mmol/L)	Control	5.1	3.0	4.0	2.4	4.0	3.1	3.0	4.6	2.7	4.5	2.2	3.8
	Dex	1.6	5.6	1.5	5.6	1.9	6.0	1.5	6.4	2.4	6.4	3.1	5.3

IAP, intra-abdominal pressure; RR, respiratory rate; EtCO_2_, end-tidal carbon dioxide; TV, tidal volume; MV, minute volume; pH, potential hydrogen; PaO_2_, partial pressure of oxygen; PaCO_2_, partial pressure of carbon dioxide; HCO_3_^−^, bicarbonate; BE, base excess; SD, standard deviation. In respiratory and blood gas variables, no significant differences are observed between different IAPs and between the control and dexmedetomidine group.

## Data Availability

All date included in the paper.

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
