# Peer review of "Cardiovascular and Respiratory Effects of Increased Intra-Abdominal Pressure with and without Dexmedetomidine in Anesthetized Dogs"

_vetsci, 2023, doi:10.3390/vetsci10110634_

Round 1

Reviewer 1 Report

Comments and Suggestions for Authors

Dear Authors,

thank you for submitting this manuscript. You will find my comments below

Author Response

Response to Reviewer 1 Comments

Point 1: Should a proper power study been undertaken before starting the experiments?

Response 1:

Response to reviewer: “ Prior to ansewering, we appreciate the commnets, power study could not be done due to the limitaion on experimental animal number.”

Point 2: Absract -reference is made to "carbonic anhydrase insufflation"--NOT applicable.

Response 2: Related contents were revised according to reviewer’s comments.

 “Five dogs were anesthetized, instrumented, and the Veress needle was equipped to adjust the IAP using a carbon dioxide insufflator.”

Point 3: Introduction    3rd para started with an abbreviation (CO). there is also reference to "patients" this should not apply to animals but ONLY humans as per the New Oxford Dictionary

Response 3: Related contents were revised and removed according to reviewr’s comments.

Cardiac output is frequently measured during anesthesia or in critical care medicine, and CO measurement allows early detection of hemodynamic instability [12]”.

Point 4: Experimental design--the end sentence is confusing as it literally refers to catheters being returned to their housing facility.

Response 4: Related contents were revised according to reviewer’s comments.

“All catheters were removed, and the dogs have been returned to their housing facility.”

Point 5: The last para referes to  "dogs being administered"   --this is incorrect only grugs are administered

Response 5: Related contents were revised according to reviewer’s comments.

 “The dogs that were not received dexmedetomidine were assigned as the control group and those received dexmedetomidine were assigned as the Dex group.”

 Point 6: The paragraph on Cardiac out put is superfluous and can be omitteed

Response 6:

Response to reviewer : “ First of all, thank you for pointing this out, as our study first introduced PulsCO method which calibrate with echocardiogram, the author believed that this background knowledge was necessary”

Reviewer 2 Report

Comments and Suggestions for Authors

This is an intersting study on the effects on the cardiac and respiratorty sytems in anaesthetised dogs produced by abdominal insufflation with carbon dioxide. .The study goes on to assess the effects of dexmetomodine on these responses

Re numbers ? Should a proper power study been undertaken before starting the experiments?

Comments on the Quality of English Language

The English will need deiting as a number of problems occur in the text

Absract -reference is made to "carbonic anhydrase insufflation"--NOT applicable.

Introduction    3rd para started with an abbreviation (CO). there is also reference to "patients" this should not apply to animals but ONLY humans as per the New Oxford Dictionary

2.5 Experimental design--the end sentence is confusing as it literally refers to catheters being returned to their housing facility.

The last para referes to  "dogs being administered"   --this is incorrect only grugs are administered

This also occurs in the discussion. The paragraph on Cardiac out put is superfluous and can be omitteed

Author Response

Response to Reviewer 2 Comments

Point 1: Simple summer: Endoscopic or laparoscopic? Advantages of minimal invasive approach. However, this technique produces capnoperitoneum.
I would suggest ot be more precise, monitoring parameters which one? You will state later. I would rewrite and reformulate the entire summery.
And are you sure that in human medicine there are none described? there are quite a good amount published in human medicine. So you should say that in veterinary literature there are no studies reporting the effect of intrabdominal increased pressure during laparoscopic surgery. And which surgery?
You got everything in the paper this simple summer do not reflect the value and quality of the study you must rewrite to show better your work and give it value!
Abstract: which parameters? How? You must say this in the abstract.

 Response 1: Related contents were revised according to reviewer’s comments.

Laparoscopic procedures have been gaining popularity in veterinary medicine, however, there are no studies evaluating the safety of using dexmedetomidine as adjuvants during laparoscopy in veterinary literature. High intra-abdominal pressure and dexmedetomidine might have deleterious cardiorespiratory effects, therefore this study was conducted to evaluate the effects of intra-abdominal pressure and dexmedetomidine on cardiorespiratory variables of healthy dogs. Five healthy beagle dogs were enrolled in the study which was conducted with crossover design. Cardiovascular and respiratory variables were monitored at different intra-abdominal pressures by inducing capnopertioneum, after washout period, same protocols were conducted with dexmedetomidine administration. Our study revealed that no significant cardiorespiratory effects were observed until intra-abdominal pressure of 20 mmHg, moreover also during administration of dexmedetomidine. These findings have shown that dexmedetomidine infusion administration may be applicable in laparoscopic procedures in healthy dogs.”

Point 2: Intrabdominal increased pressure not only create cv effects but also intraparenchymal problems. You did not calculate perfusion of kidneys of intragastric or transdiaphragmatic pressures, so it is important to discuss that this study was looking only to the cardiovascular and respiratory effects and other effects were not evaluated. 20 mmHg it is an high values and in literature has been reported that operators should stay below 12 mmHg to limit organ issues and intraparenchymal perfusion problems.

Response 2: Related contents were revised according to reviewer’s comments.

According to previous studies, intra-abdominal hypertension not only influences cardiorespiratory effects but also alters regional blood flow and impairs tissue perfusion, resulting in organ dysfunction [1,4]. However, since our aim was not to evaluate organ perfusion, we only look in to the cardiovascular and respiratory effects.”

Point 3: Dexmedetomidine could have high beneficial effects on organ protection against perfusion. This neuroprotective, kidney injuries and this is then very interesting the possible positive effects of the use in laparoscopic procedures. This can be discussed to improve the impact and quality of the paper.

Response 3:

Response to reviewer: “ Prior to ansewering, we appreciate the reviewer’s commnets. As our study did not evaluate the organ perfusion, we would like to further evaluate these positive effects of dexmedetomidine and discuss in next study.”

Point 4: 2.2. the endotracheal tube was connected to the y or coaxial branches of the anaesthesia circle circuit (...) and maintained with iso in oxygen 100%.

Response 4: Related contents were revised according to reviewer’s comments.

“After endotracheal intubation, each dog was positioned in dorsal recumbency, and anesthesia was maintained with isoflurane (Isoflurane; Hana Pharm, Korea) in 100% oxygen through a circle rebreathing system (9100c NXT; GE Healthcare, China).”

Point 5: For cardiac output in the keyword, you wrote PulseCO and in the text LIDCO, but you measured the CO by echocardiography? There is a bit of confusion in the indication described. PulseCO is the the technology used in the LIDCOTM I think you should add this information in the text. It is important to report this not only in the discussion section but also in the material and methods.

Response 5: Related contents were revised according to reviewer’s comments.

This LiDCO™ monitor is utilizing PulseCO method, not the LiDCO system.”

Point 6: Did you used the values data from LIDCO or from the ultrasound measurements?

Response 6:

Response to reviewer: “In the present study, both data were used. Ultrasound measurements were used as calibration only, after calibration, LIDCO monitor data were recorded as cardiac output.”

Point 7: This was a non-randomized, cross over approach study all dogs were first anaesthetized without dexmedetomidine and after 1 week all dogs received dexmedetomidine. Why didn’t you decide to do it in a randomized way? Don’t you think it could give bias in the experiment?

Response 7:

Response to reviewer: “In the present study, mean and median data of all dogs were compared. Moreover, it should be noted that our study did not include subjective evaluation, therefore, was not designed as randomized trial.”

Point 8: I think that all the dogs should have been ventilated. The IPPV will influence the CO and if you do not standardize the major variable your results are biased.
Because all your dogs required IPPV we can read the results, but you should than say at which timepoint you started IPPV and see if there was a change in CO before and after and discuss it

Response 8:

Response to reviewer: “We appreciate the reviewer’s comment. Since our study planned to evaluate the respiratory function of healthy dog with intra-abdomnial hypertension, we hope to see the differences in respiratory variables between different IAP without ventilation. However, one dog showed severe hypercapnia and intermittent positive pressure ventilation was started in this dog alone. Moreover, IPPV performed irregularly and hypercapnia was not resolved despite of IPPV. Therefore, the effects of IPPV in the present study cannot be evaluated.”

Point 9: “As the PulseCO method is developed for measurements in human medicine, one-point calibration is necessary, andthese studies calibrated the monitor with CO measured using the lithium-dilution or thermodilution method [13,22,23].” True...” However, in veterinary clinical settings, there is no accepted method for measuring CO.” NOT TRUE!!!the same is in veterinary medicine!
Lithium measurement are commonly used, true that are relatively invasive and pulse counter techniques are more friendly used and less invasive but please reformulate and review more literature. It is nice you calibration using an ultrasound based approach. Has this methods been described and validated before? Can you add some references? there are studies as you mention afterwords that foud differences between those technology so I don’t understand how you decided to calibrate this according to ultrasound but I am really interested to hear your reason why

Response 9: Related contents were removed according to reviewer’s commnets.

” However, in veterinary clinical settings, there is no accepted method for measuring CO.”

Response to reviewer: “We appreciate the reviewer’s comments, in human ECC, LiDCO monitors have recently been introduced, and the data from LiDCO are being calculated using the patient’s body surface area. However, animal cannot be applied to these monitor as it is only setted for human. To apply this to animals, cardiac output calculation is necessary for the calibration. In the present study, we decided to calculate cardiac output using echocardiogram to make applicapable in veterinary clinical settings beacause calibration with invasive methods(lidco or pactd) might worsen the critical patient. There is no reference of this method so we are planning to further evaluate this LiDCO monitor.”

Point 10: I think it is also a must to say that the CO2 insufflated in the abdomen is eliminated toward the lungs, so hypercapnia is an very expected problem during laparoscopic procedures. We must be always sure to have our patients intubated and ventilation is strongly suggested in this cases to maintain normocapnia and avoid respiratory acidosis

Response 10: Related contents were revised according to reviewer’s comments.

“While the causes of hypercapnia can be diverse, in this study, it is believed that respiratory suppression due to anesthetics and elimination of carbon dioxide through the lungs during capnoperitoneum were the main factors leading to hypercapnia rather than increased IAP [26-28].”

“In conclusion, ventilation during laparoscopic procedures is recommended to maintain normocapnia and avoid respiratory acidosis.”

Point 11: Why you say in the limitation that this is a cross over approach??? Where is the cross over approach applied?

Response 11:

Response to reviewer: “First of all, we appreciate reviewer’s comments, in the present study, crossover study was adapted that all dogs first anesthetized without dexmedetomidine and, after one week of washout period, all dogs received dexmedtomidine with similar experiments protocols. In limitation, we comment’s this crossover design to explain that for purpose of minimizing the influence of small number of animal”

Round 2

Reviewer 1 Report

Comments and Suggestions for Authors

Dear Authors,

thank you for the changes made I think the manuscript is now ready for publication

thank you again for the work done!

BW

Author Response

Response to Reviewer 1 Comments

Because there were concerns about the accuracy of the English language in the review, we had a professional agency perform comprehensive English editing of the paper.

Point 1: on page 3. I think that pH is a stand-alone term well enough known to not need any definition. If you wish to use a definition, then I think that "pH" is better defined (if desired or necessary) as "the logarithm of the reciprocal of hydrogen ion concentration in grams per liter." The definition of 'pH' as "the potential OF hydrogen" I think is somewhat  archeic, and the expression you have used "potential hyrdogen" is known to me only as a lay-language term.

Response 1: Related contents were revised according to reviewer’s comments.

Potential hydrogen(pH) -> pH

Point 2: on page 2, "However, in veterinary medicine, the evaluation of the hemodynamic effects of dexmedetomidine during abdominal insufflation has not been studied." After removing adverbial phrases and adjectives, the true meaning of this sentence, literally, is = "The evaluation has not been studied." It would be easier for a reader were this to be reworded as "However, in veterinary medicine the hemodynamic effects of dexmedetomidine during abdominal insufflation with CO2 have not been studied."

Response 2: Related contents were revised according to reviewer’s comments.

 “However, in veterinary medicine, the hemodynamic effects of dexmedetomidine during abdominal insufflation with CO2 have not been thoroughly investigated.”

Point 3: It is not appropriate to discuss your "belief" where that bias was not demonstrated. Where there are literature references that support your preferred explanation, then certainly discuss and cite them as you have, but you need to explain why your results do or do not not support that bias, and whether there is contrary evidence elsewhere in the literature - i.e. present a balanced and complete discussion argument, please (particularly relating to the ETCO2 vs PaCO2 and the potential effect on monitored variables).

Response 3: Related contents were revised and removed according to reviewr’s comments.

“Previous studies have demonstrated that respiratory disturbances associated with elevated IAPs can occur in pediatric human patients as well as animals [20,25]. These effects are thought to be triggered by the cranial displacement of the diaphragm, leading to a reduction in functional residual capacity, residual volume, and total lung capacity [1]. Richardson et al. indicated that significant respiratory effects were observed at IAPs exceeding 25 mmHg, while our findings similarly indicate that no significant differences in respiratory variables were observed until IAP reached 20 mmHg [20]. However, although our results were not statistically significant, PaCO2 increased following abdominal insufflation, peaked at T3 and T4 in the control and Dex groups, respectively, and decreased following desufflation. These results suggest that IAPs below 20 mmHg induced by capnoperitoenum affect respiratory function but with a minor effect.”.

“Lined sentences have been removed to support that intra-abdominal pressure does not affects on respiratory function. ”

Point 4: In reporting the study design, it is important that a reader could replicate the study exactly. So, there are a couple of missing details. For example, what is the duration between T0, T1, T2 etc, i.e. how long did it take to adjust the IAP in each animal and what was the variability of those times? Did the inhalant anaesthetic need adjustment to maintain depth of anaesthesia? If so, between what limits and how frequently - and could that have impacted SBP or DBP? Sometimes I find that reporting study designs for anaesthetic studies is best achieved efficiently using figures. I think your paper wold benefit from one or two illustrations that helped convey the detail of the study design.

Response 4:

Response to reviewer : “ In the present study, the duration between timepoints were 10 minutes and we have designated and fixed this time as a stabilization period. After 10 minutes of stabilization, aforementioned data was collected. “

“Inhalant anesthetics have significant effects on cardiovascular system, and our study aim was to monior cardiovascular effects of intra-abdominal pressure. Therfore, inhalant anesthetics were not adjusted based on the depth of anesthesia and fixed concentration of isoflurane using MAC in order to minimize the variability of study.“

“We have inserted study timeline and illustrated in Figure 1.”
